# Copper-Chitosan Nanocomposite Hydrogels Against Aflatoxigenic *Aspergillus flavus* from Dairy Cattle Feed

**DOI:** 10.3390/jof6030112

**Published:** 2020-07-21

**Authors:** Kamel A. Abd-Elsalam, Mousa A. Alghuthaymi, Ashwag Shami, Margarita S. Rubina, Sergey S. Abramchuk, Eleonora V. Shtykova, Alexander Yu. Vasil’kov

**Affiliations:** 1Plant Pathology Research Institute, Agricultural Research Center (ARC), 9-Gamaa St., Giza 12619, Egypt; 2Biology Department, Science and Humanities College, Shaqra University, Alquwayiyah 19245, Saudi Arabia; malghuthaymi@su.edu.sa; 3Biology Department, College of Sciences, Princess Nourah bint Abdulrahman University, Riyadh 11543, Saudi Arabia; 4A.N. Nesmeyanov Institute of Organoelement compounds (INEOS) of Russian Academy of 13 Sciences, 119454 Moscow, Russia; margorubina@yandex.ru (M.S.R.); abr@polly.phys.msu.ru (S.S.A.); alexandervasilkov@yandex.ru (A.Y.V.); 5V. Shubnikov Institute of Crystallography of Federal Scientific Research Centre “Crystallography and Photonics” of Russian Academy of Sciences, 119333 Moscow, Russia; shtykova@ns.crys.ras.ru

**Keywords:** aflatoxins, *Aspergillus* section *Flavi*, chitosan, feeds, nanocomposites

## Abstract

The integration of copper nanoparticles as antifungal agents in polymeric matrices to produce copper polymer nanocomposites has shown excellent results in preventing the growth of a wide variety of toxigenic fungi. Copper-chitosan nanocomposite-based chitosan hydrogels (Cu-Chit/NCs hydrogel) were prepared using a metal vapor synthesis (MVS) and the resulting samples were described by transmission electron microscopy (TEM), X-ray fluorescence analysis (XRF), and small-angle X-ray scattering (SAXS). Aflatoxin-producing medium and VICAM aflatoxins tests were applied to evaluate their ability to produce aflatoxins through various strains of *Aspergillus flavus* associated with peanut meal and cotton seeds. Aflatoxin production capacity in four fungal media outlets revealed that 13 tested isolates were capable of producing both aflatoxin B1 and B2. Only 2 *A. flavus* isolates (Af11 and Af 20) fluoresced under UV light in the *A. flavus* and *parasiticus* Agar (AFPA) medium. PCR was completed using two specific primers targeting aflP and *aflA* genes involved in the synthetic track of aflatoxin. Nevertheless, the existence of *aflP* and *aflA* genes indicated some correlation with the development of aflatoxin. A unique DNA fragment of the expected 236 bp and 412 bp bands for *aflP* and *aflA* genes in *A. flavus* isolates, although non-PCR fragments have been observed in many other Aspergillus species. This study shows the antifungal activity of Cu-Chit/NCs hydrogels against aflatoxigenic strains of *A. flavus*. Our results reveal that the antifungal activity of nanocomposites in vitro can be effective depending on the type of fungal strain and nanocomposite concentration. SDS-PAGE and native proteins explain the apparent response of cellular proteins in the presence of Cu-Chit/NCs hydrogels. *A. flavus* treated with a high concentration of Cu-Chit/NCs hydrogels that can decrease or produce certain types of proteins. Cu-Chit/NCs hydrogel decreases the effect of G6DP isozyme while not affecting the activity of peroxidase isozymes in tested isolates. Additionally, microscopic measurements of scanning electron microscopy (SEM) showed damage to the fungal cell membranes. Cu-Chit/NC_S_ hydrogel is an innovative nano-biopesticide produced by MVS is employed in food and feed to induce plant defense against toxigenic fungi.

## 1. Introduction

The contamination of agricultural and dairy products with aflatoxins is a major problem for economic and public health. Aflatoxins (AFs) are fungal subsidiary products mainly developed by *Aspergillus flavus* and *Aspergillus parasiticus* strains on cereals, nuts, dried fruits, dairy, and animal feed under warm and humid conditions [1,2]. The significant source of AFs spoilage is *A. flavus*, especially aflatoxin B1, which has received a lot of attention in the food and feed industry [3]. High concentrations of aflatoxin could even prompt the disease of aflatoxicosis, an infection that affects serious disease and can lead to cancer in severe cases [1,4,5]. Additionally, chronic absorption of aflatoxins causes various adverse effects, such as increased susceptibility to various pathogens, loss of production, and a decrease in milk production yield and quality in dairy cattle [6]. The nanotechnology approach seems to be an encouraging, effective, and affordable way to reduce the health problems of mycotoxins in humans and animals. There are three different approaches to reduce mycotoxin risks: effects on mold and flour retention, mycotoxin, and minimization of toxic effects by various nanomaterials [7,8,9]. Chitosan and self-assembled benzoic acid polymers were synthesized, and it was found that the encapsulation of CS-BA nanogels significantly enhanced the half-life and antifungal activity properties of thyme oil against *A. flavus* strains [10]. The antifungal efficacy of mycogenic silver nanoparticles hybridizing with simvastatin against three species of the *Aspergillus Flavi* group was measured. Some nano-formulations regulated the development of the toxigenic *Aspergillus* species [11]. Plant-mediated CuO NPs were synthesized from *Cissus quadrangularis* and applied as antifungal agents against *A. niger* and *A. flavus*. The produced nano-copper showed a better performance than the carbendazim fungicide [12]. In addition, hybrid nanocomposites based on organic polymeric and inorganic matrices as effective anti-aflatoxigenic strains were explored [3,4,13,14]. Chitosan-based nanocomposite film vapor assays were applied to hybrids between thyme-organo, thyme-tea tree, and thyme-peppermint EO mixtures and demonstrated strong antifungal action against some toxic fungi, including *A. flavus, A. parasiticus,* and *P. chrysogenum*, limiting their production ranged from 51 to 77% [15]. There is a direct association between the concentration of aflatoxin M1 (AFM1) in milk and aflatoxin B1 (AFB1) in dairy cattle feed which results in AFM1 being found in the milk of animals on contaminated feeds with AFB1 [16]. To our understanding, antifungal action of copper-chitosan nanocomposite-based chitosan hydrogels (Cu-Chit/NC_S_ hydrogels) against *A. flavus* strains from animal feed samples is not previously studied. Present study aimed to: (1) recognize *alfP* and *aflA* as two essential genes that lead to development of aflatoxin in animal feed via *Aspergillus* genus. (2) determination of AFB1 and AFB2 frequency and distribution of *A. flavus* strains in relation to feed delivered to dairy cows in small farms. (3) copper-chitosan nanocomposite was produced utilizing metal vapor synthesis (MVS), the physicochemical characteristics of the nanocomposites formed were described by electron microscopy (TEM) transmission, X-ray fluorescence analysis (XRF), and X-ray scattering (SAXS) small angle. (4) The fungicidal effect of the hydrogel Cu-Chit/NCs were screened against three *A. flavus* strains. (5) Protein, isozymes, and DNA fragmentations were investigated using two electrophoresis techniques, finally, scanning electron microscope was used to assess morphological changes in NCs-treated fungi.

## 2. Materials and Methods 

### 2.1. Chemicals and Reagents

High-quality Acetone with a special purity 99.5% was used as a solvent for the production of metal nanoparticles via metal vapor synthesis (MVS) technique. Prior to the synthesis solvent, it was dried under molecular sieves (4 Å) and degassed in a vacuum pump under 10^−1^ Pa by freezing and thawing cycles. The metal source was Cu foil (99.99 percent) with a surface pre-treated with concentrated HNO_3_ and diluted H_2_O to remove oxide film.

In this work, two types of chitosan were used. Chitosan with a high molecular weight (ChitHMW) was purchased from ACROS Organics (Wheaton, IL, USA). Chitosan with a low molecular weight (ChitLMW) was bought from Wirud (Hamburg, Germany). Until impregnation, the chitosan powder was degassed at 40 °C for 12 h under a vacuum of 10^−1^ Pa. Oxalic acid dihydrate was of analytical consistency.

### 2.2. Preparation of Chitosan Powder Modified with Cu NPs

Chitosan powder decorated with Cu NPs was prepared according to the procedure described in the previously published works [17,18,19,20]. For the preparation of organosol, 0.56 g of Cu foil (about 200 µm of thickness) was resistively dispersed from the tantalum boat (90 mm × 5 mm) and co-condensed with 160 mL of acetone on the liquid nitrogen-cooled walls of a quartz 5 L vessel. This procedure was carried out at a residual pressure of 10^−5^ within 1 h. Then the cooling was removed and the reactor was filled with pure argon. Under these conditions, the cryomatrice warmed to room temperature naturally within 15 min. As a result, the Cu-acetone organosol was obtained. The calculated solvent-to-metal molar ratio in the synthesis was 250:1 and the concentration of the copper organosol was 5 10^−2^ M. Cu-acetone organosol was then infiltrated with chitosan powder (HMW or LMW) in an evacuated Schlenk vessel. During the deposition procedure, the flask was stirred manually to obtain homogeneous material. Thereafter, the solvent was removed and the chitosan powder containing Cu NPs was dried in a vacuum of 10^−1^ Pa at 40 °C for 6 h. As a result, two types of powdered Cu-carrying composites based on ChitHMW and ChitLMW were prepared.

### 2.3. Preparation of Chitosan Gels Modified with Cu NPs

The following technique was used to prepare chitosan gels filled with Cu NPs. Cu-carrying chitosan powder (1.32 g, 5 percent *w*/*w*) was dissolved in an oxalic acid solution (25 mL, 1 M) with vigorous stirring at 80 °C for 30 min. The mixture was poured into the cylindrical molds (20 mm × 10 mm) and put in the water bath below 22 °C to maintain a steady gel formation temperature. Each mold had 4 g of chitosan solution. After 12 h, the prepared gels were soaked in a beaker filled with distilled water and cleaned thoroughly from excess acid until neutral pH was reached. With the methods described above, two types of gels were prepared:

Cu@ChitHMW—chitosan hydrogel doped with Cu NPs based on ChitHMW

Cu@ChitLMW—chitosan hydrogel doped with Cu NPs based on ChitLMW

For best solidification, chitosan gels were stored in a water/isopropanol (6:1, v/v) bath at room temperature (RT).

### 2.4. Characterization Techniques

#### 2.4.1. Transmission Electron Microscope (TEM)

TEM images were performed with a transmission electron microscope LEO 912AB OMEGA, Zeiss (Oberkochen, Germany) at an acceleration voltage of 100 kV. Cu-carrying chitosan composites for measurements were previously suspended in deionized water (resisting 18 MΩ) and sonicated in an ultrasonic bath for 15 min at RT. Then, a small drop of the suspension was dripped onto a copper grid (200 mesh) previously coated with formvar film. Then, the samples were dried at RT for 15 min and placed under the microscope.

#### 2.4.2. X-Ray Fluorescence (XRF) Analysis

To determine metal concentration (% *w*/*w*) in the composites, a VRA 30 X-Ray fluorescent analyzer (Leipzig, Germany) was used. To excite XF, an X-Ray tube with a Mo anode was used at the acceleration of 50 kV and current of 20 µA. For analysis, Cu-chitosan powders or gels in an amount of 10–12 mg was thoroughly ground and pressed into pills. Then the XRF spectra of the composites and reference samples were recorded. The standard buffer solution for spectrometer calibration was composed of a mixture of polysterene/metal salt. Quantity analysis was conducted through comparison of the peak intensity of Cu Kα line in XRF spectrum of the composite with the values of the calibration curve obtained previously. 

#### 2.4.3. Conventional Small-Angle X-Ray Scattering (SAXS) Analysis

SAXS measurements were done on laboratory diffractometer “AMUR-K” (developed in A. V. Shubnikov Institute of Crystallography, Moscow [21]). Wavelength of X-rays *λ* = 0.1542 nm was used, applying Kratky type geometry covered the range of scattering vector modulus 0.12 < *s* < 6.0 nm^−1^ (*s* = 4*πsinθ*/*λ*; 2*θ* is the scattering angle). Experimental data were normalized to the intensity of the incident beam, and then a correction on collimation error was made according to standard procedure [22]. Further data processing and interpretation was done using the program suit ATSAS [23].

Volume size distribution functions *D_V_(R)* of heterogeneities in the specimens and distance distribution functions *p(r)* were computed by means of the regularization technique realized in program GNOM [24]. The low-resolution shapes of the Cu nanoparticles in the Cu-carrying chitosan were reconstructed ab initio using distance distribution function *p(r)* and program DAMMIN [25]. The program utilizes a simulated annealing algorithm to build models fitting the experimental data *I_exp_(s)* that minimizes the discrepancy:$$\chi^{2}=\frac{1}{N-1}{\sum_{j}\left[\frac{I_{\exp}\left(s_{j}\right)-cI_{calc}\left(s_{j}\right)}{\sigma \left(s_{j}\right)}\right]}^{2}$$
where *N* is the number of experimental points, *c* is a scaling factor and *I_calc_(s_j_)* and *σ(s_j_)* are the calculated intensity from the model and the experimental error on the intensities at the momentum transfer *s_j_*, respectively. The program DAMMIN was run of about a dozen separate calculations to identify the most typical models.

### 2.5. Aflatoxin-Producing Ability Medium 

Selected isolates from *A. flavus* (21 isolates) associated with dairy cattle feed samples collected from different governorates in Egypt were screened for aflatoxins production by four solid media. Four culture media recipes including *A. flavus* and *parasiticus* Agar (AFPA) (20 g/L yeast extract; 10 g/L bacteriological peptone; 0.5 g/L ferric ammonium citrate; and 15 g/L agar) [26], coconut agar (CA), PDA+ 20% NaCl, and PDA were employed to confirm aflatoxins production. All tested media were equipped as described earlier by some mycologists [1,27]. Aflatoxins production was detected via UV fluorescent light, light brown circle closed to fungal colonies was appeared after one-week incubation at 28 °C.

### 2.6. VICAM Aflatoxins Assay

Culture broth filtrates of 21 aflatoxigenic *Aspergilii* collected from peanut cake and cotton seeds were analyzed for aflatoxins production (B1, B2, G1, and G2) (Figure 1) using AflaTest Immunoaffinity (VICAM) Chromatography assay that quantifies total aflatoxin concentrations according to the manufacturer’s instructions. Briefly, 5 mL of diluted sample extract (2:3, extract:water) was filtered through the column with immunological properties with a drop per second. The column was cleaned with 10 mL of water by running two drops per second through the column. Aflatoxin was eluted by transferring 1 mL of methanol at one drop per second from the column. A volume of 1 mL of bromine developer was added to the methanol elute and the total aflatoxin concentration was read in a pre-calibrated VICAM Series-4 Fluorometer set at 360 nm absorption and 450 nm emissions with a detection limit of 2 ppb (2μg/kg). Results for each sample were averaged and reported in ppb [28].

### 2.7. DNA Extraction

For the PCR amplification and DNA degradation assays, *Aspergillus* mycelium was grown in 20 mL of potato dextrose broth liquid medium (24 g/L of potato dextrose broth (Difco Laboratories, Detroit, ML, USA)). Fungal mats were gathered by separation through mesh sieves (40 mm), finally washed using sterile deionized, and dropped inside a Whatman filter paper to eliminate extra water. For homogenization, fungal mycelium was milled to acceptable powder in a mortar employing liquid nitrogen. The DNA protocol modified by Bahkali et al. [29] was used to obtain a highly purified DNA amplicon.

### 2.8. PCR Assay for A. flavus Detection

For specific detection of *A. flavus*, 64 PCR reaction contained 2 µL of the extracted DNA and 23 µm PCR mix containing 11.5 µL Taq DNA polymerase (Jena, Germany), 5 mM of two specific primers for *A. flavus aflP* (F-5′-CATGCTCCATCATGGTGACT-3′), (R-5′ CCGCCGCTTTGATCTAGG-3′) [30], *aflA* (F-5′-GGTGGT GAAGAAGTCTATCTAAGG-3′), and (R-5′AAGGCATAAAGGGTGTGGAG-3′) [31]. PCR thermal cycler program was adjusted as follows: 7 min at 94 °C tracked by 40 amplification cycles at 94 °C for 30 s, annealing temperature 62 for 30 s, 72 °C for 30 s, and then 72 °C for 3 min for the final extension. The amplified DNA was separated via 2% agarose gel electrophoresis containing ethidium bromide at 90 V for 30 min. Agarose gel was detected in UV transilluminator light via Gel Documentation System (Uvitec, Cambridge, UK).

### 2.9. Antifungal Assay

To determine the inhibition of mycelial growth of *A. flavus*, four Cu-Chit/NCs gel concentrations (60, 120, 180, and 240 ppm) were prepared as hydrogel discs for every disc containing 60 ppm. The anti-fungal efficacy of Ag-Chit-NCs was assessed by determining the reduction in fungal growth of *A. flavus* using agar-well diffusion assessment [32]. Each concentration of Cu-Chit/NCs gel has been applied to PDA plate, and petri dishes were inoculated with *A. flavus* fungal disks. Flavus isolates were incubated at 28 °C for 10 days. The inhibition factor of growth was estimated and evaluated by the equation below. [33]. Growth Inhibition (percent) = (R1−R2)/R1 × 100 where R1 was the control’s radial growth and R2 for each therapy was the radial growth. After one week, the photographic record and the development of the radial colony was calculated. The experiments were conducted in three folds.

### 2.10. Protein Profile Degradation Assay

To investigate the Cu-Chit/NCs gel mediated protein expression in *A. flavus,* SDS-PAGE analysis was performed by Laemmli method [34]. The extracted protein from the fungal mycelium was treated with 180 ppm of Cu-Chit/NCs gel concentrations and incubated for 8 h. SDS-PAGE was performed using a 5–10% gradient of polyacrylamide gels containing 0.1% SDS. Proteins were investigated in 1.5 mm and 15 cm gels that work in dual vertical electrophoresis glass plates (Hoefer Scientific Instruments, San Francisco, CA, USA). Twenty microliters of the extracted protein were inoculated into polyacrylamide gels. SDS-PAGE samples were differentiated by separating the acrylamide gel at a stable electrical current of 30 mA, and by using a separate gel at room temperature at 90 mA, the gel was stained with silver staining. [34]. The typical molecular weight used for gel analysis was the Sigma protein marker, which is between 66,000, 45,000, and 22,000 kDa.

### 2.11. Native PAGE Isozyme Assay

Fungal isozymes were purified by grinding 100 mg of fungal mats in 1.0 mL extraction buffer (0.1 M Tris-HCl + 2 mM EDTA, pH 7.8). The extracted enzyme from fungal mats was treated with 180 ppm of Cu-Chit/NCs gel concentrations and incubated for 8 h. Native-PAGE was used to separate two enzymatic activities under native conditions [35]. Electrophoretic technique were conducted with the electrode buffer Tris/Glycine (pH 8.3) using 5 percent of the stacking gel and 6 percent of the separating gels. The 5 μL enzyme samples were placed over each well of the stacking gel and the gel was initially run at 60 V replaced by 100 V later. After running native acrylamide gel, and for staining glucose 6-phosphate dehydrogenase (G6PD) (EC.1.1.1.49), native protein gel was incubated in a staining solution (0.1 mM tris-HCL buffer, pH8.8, 7.5 glucose 6-phosphate (di-sodium salt), 20 mg NADP, 10 mg MTT, 10 mg PMS, 0.2 M MgCl_2_) in the dark at 37 °C until dark blue band appear. To stop the reactions, the isozyme gel was washed and fixed in 50% ethanol [36]. Peroxidase isozyme (EC 1.11. 1.7) was stained with incubated gels in a staining solution (50 mM phosphate buffer, pH5.0, 50 mg benzidine dihydrochloride, and 3% hydrogen peroxide), and the gel was washed in water and fixed in 50% glycerol [37].

### 2.12. Binding/Degradation of Genomic Fungal DNA

To check DNA quality, 10 microliters of *A. flavus* DNA were treated with Cu-Chit/NCs hydrogel (180 ppm) for 2 h at 37 °C. The DNA amplicon treated NCs gel was separated on 1.5% (*w*/*v*) agarose gels prepared in 1× Tris-acetate-EDTA (TAE) and stained with ethidium bromide (EtBr, 10 mg/mL). Five microliters of extracted and treated DNA from every pattern, along with 1 μL DNA loading dye, become loaded into the wells. Agarose gel was run for 30 min at 90 V and visualized to check for DNA degradation inside the GelDoc (Uvitec, Cambridge, UK).

### 2.13. Scanning Electron Microscopy (SEM)

Agar disks of *A. flavus* had been inserted into sterile cellophane films and transferred to PDA mixed with 180 ppm of Cu-Chit/NCs hydrogel in Petri dishes, with one plugin keeping in the middle of the plate, and 3 replicates were used for every strain. The Petri dishes were incubated at 28 °C for 7 days. Cellophane coated with *A. flavus* was transferred and stuck in 2% (*w*/*v*) glutaraldehyde at 4 °C for 12 h in 0.1 M phosphate buffer (PB; pH 7.0). Aspergillus specimens will be put in the desiccator before further use. Upon drying, the samples prepared are assembled into the Poutron SEM coating system using standard double-sided adhesives with ½ inch SEM nozzles and with a gold-palladium gold coating (60 s, 1.8 mA, 2.4 kV). All samples were tested with JEOL JXA-480 SEM (JEOL, Tokyo, Japan) at the National Research Center in Giza, Egypt.

## 3. Results

### 3.1. Preparation and Characterization of Cu-Carrying Chitosan Powders

Chitosan modified with metal NPs was prepared in two steps. Firstly, Cu NPs were prepared by interaction of metal vapors with acetone vapors according to the MVS protocol (see experimental part). During the second step, freshly prepared Cu-acetone organosol was deposited in situ onto chitosan powders. During the deposition procedure, the flask was stirred manually to obtain homogeneous material. Discoloration of organosol indicated the completeness of the nanoparticle’s deposition on the biopolymer support and the color of powders changed from beige to dark-green. XRF analysis shows that the metal proportion of the Cu-carrying powders based on ChitLMW and ChitHMW is 0.5 and 0.83%, respectively. Previously, the same method for preparing Cu-carrying chitosan powders with a high copper concentration of 3–5% *w*/*w* was used [20].

The TEM images in the bright field and the selected area diffraction pattern (SAED) for the newly prepared Cu-acetone organosol have been seen in Figure 2A. As can be shown, SAED has diffuse reflexes suggesting the production of a significant number of very small particles. NPs have a mainly spherical shape and blurry boundaries. The NPs’ sizes estimated from Figure 2B are in the range of 1 < d < 4 nm. The good solvating properties of acetone for preparing Cu NPs were shown in previously published works [38,39].

Five rings correspond to the lattice planes of Cu and Cu_2_O. As a result of the proximity of some interplanar distances of Cu and Cu_2_O and relatively broad rings, their superposition was observed. The formation of a core-shell structure of copper NPs with metallic copper as a core and copper oxide (I) as a shell can be assumed. A similar structure of Cu NPs in organosols prepared with different solvents via MVS was detected [40].

In Figure 3 TEM images in bright/dark field and SAED of powdered chitosan doped with Cu NPs using the impregnation step are shown. It was detected that Cu-carrying chitosan composite contains Cu and Cu_2_O phases as well as Cu-acetone organosol (Figure 3B). The crystallite sizes estimated from dark field image are in the range of 2–4 nm (Figure 3D).

It was previously demonstrated that the surface of the composite prepared with Cu-acetone organosol contains two oxidized copper states Cu^2+^ and Cu^+^, with concentrations (at. %) of 10.7 and 3.6%, respectively [20]. Experimental SAXS curves of Cu-carrying chitosan and pristine non-modified chitosan (ChitLMW) are described in Figure 4A. 

Volume size distribution functions *D_V_(R)* of heterogeneities presented in pristine non-modified chitosan and that of Cu nanoparticles embedded in the chitosan are shown in Figure 4B. To obtain *D_V_(R)* only for the Cu nanoparticles, a difference SAXS curve was calculated by subtraction of the scattering of the pristine non-modified chitosan from the scattering of the Cu-carrying chitosan (Figure 4A insert). As we can see from Figure 4B, Cu nanoparticles in this system are practically monodisperse and more compact (average size is about 1.5–2 nm) to compare with the sizes of the pores of the pristine non-modified chitosan (about 3 nm), where the Cu nanoparticles are located. A small detectable number of larger particles (possible aggregates or clusters of Cu nanoparticles) is also present in the sample. Due to the practically monodisperse character of the *D_V_(R)* function for Cu nanoparticles, one can reconstruct an average shape of the Cu nanoparticles [21]. For the ab initio restoration by the program DAMMIN [24], distance distribution function *p*(r) was calculated. A shortened curve with no initial part at the range of momentum transfer < 0.7 nm^−1^ was used for the calculation to minimize the influence of scattering from large aggregates on the result of the shape restoration. The distance distribution function *p(r)* is shown in Figure 5 (insert on the top right) along with a model scattering curve from a restored shape and with a smoothed curve after the of collimation corrections.

The restored shape of the Cu nanoparticles is a cluster consisting of 5–6 individual Cu nanoparticles with the average sizes of about 1.5–2.0 nm and with the length of the cluster of about 5 nm. Due to the presence of some amount of large aggregates it is impossible to restore the shape of the individual Cu nanoparticles. However, the shape of the cluster is restored with very good accuracy: *χ^2^ =* 0.92, and the separate nanoparticles in the cluster are clearly visible. 

Chitosan gels have been produced by ionic physical gelation of the oxalic acid-biopolymer [41,42]. Two types of Cu-carrying chitosan gels were obtained with the consecutive procedures of dissolution in oxalic acid at high temperature, gelation and thorough washing procedures (Figure 6).

### 3.2. Aflatoxins Production Ability

The production ability of toxigenic isolates was screened on four solid media. Beige rings seen without light are observed in aflatoxigenic fungal cultures. It is also possible to visualize the blue fluorescence ring that surrounds the aflatoxigenic colony under ultraviolet light. The aflatoxigenic *A. flavus* °C. The detected beige ring diameter and the strength of its fluorescence emission were improved under UV over time with the maximum observation by the end of the week (Table 1). It is possible to use AFPA media that is suitable for fast screen aflatoxigenic fungi associated with feeds. Four forms of aflatoxin activity have been quantified and measured (B1, B2, G1, and G2). Production patterns of AFs by aflatoxigenic *A. flavus* isolates are presented in Table 1. Thirteen of these were producers of AFB_1_ and AFB_2_ aflatoxins and 8 were nonproducers of aflatoxins. Thirteen of the isolates produced aflatoxin B1 ranging from 4.50 to 19.44 ppb, while B2 was produced in the same isolates with a 0.02–5.29 ppb. *A. flavus* (Af1) produced the highest AFB1 concentration (19.44 ppb) while *A. flavus* (Af13) produced the intermediate quantity of AFB1 (9.10 ppb) and Af13 produced the lowest quantity of AFB1 (4.50 ppb). None of the tested isolates produced aflatoxins G1 and G2.

### 3.3. Aspergillus flavus PCR Detection

A total of 21 *Aspergillus flavus*, and *Aspergillus* related isolates such as two isolates for both *Aspergillus clavatus*, *Aspergillus ochraceous*, *Aspergillus niveus*, *Aspergillus terreus*, *Aspergillus fumigatus*, *Aspergillus versicolor*, *Penicillium paneum*, *Penicillium expansum*, *Penicillium citrinum*, *Penicillium verrucosum* in addition to one isolate from *Alternaria alternata* were used for testing specificity of primers. Figure 7 indicates the findings of the PCR product inspection of the agarose gel assay; the presence of a band at 236 bp for primers *aflP* and 412 bp *aflA* primers, respectively and suggested the same predicted PCR product. The samples were prepared independently from individual strains, with an identical amount of DNA (60 ng/μL). Figure 7 shows the results for screening the specificity of the tested primers. Thus, two sets of primer pairs (*aflP* and *aflA*) are specific for the detection of *A. flavus* (Figure 7A,C). The results show that only *A. flavus* DNA can be amplified (lines 1 to 21), no PCR products from other *Aspergillus* species and Penicillium were obtained from lane 1 to 21 (Figure 7B). For rapid and accurate detection of *A. flavus* isolates tested in the current report, two PCR primer sets (*aflP* and *aflA*) were employed. The applicable primers amplified a PCR fragment sited near the 18SrRNA region with 95.3% efficiency and 100% specificity. Ultimately, existing examinations suggest the powerful specificity of the *aflA* PCR primer over different typically available diagnostic primers for correct, speedy, and large-scale identification of *A. flavus* isolated from feeds. Most *A. flavus* strains were detected with the aid of specific-PCR by using the examined primer sets.

### 3.4. Antifungal Activity of NCs 

Various concentrations of Cu-Chit/NCs hydrogels were used to study the inactivation of *A. flavus* mycelia growth. The antifungal activity of the synthesized Cu/NCs chit gel was assessed by measuring the radial growth of mycelium for all treatments (Figure 8). The highest mycelial growth inhibition was found at a concentration of 240 ppm followed by 180, 120, and 60 ppm concentrations of Cu-Chit/NCs hydrogel. The antifungal activity of Cu-Chit/NCs hydrogel did not increase with increasing concentrations ranging between 60 and 120 ppm, while the inactivation rate constant increases with the concentration of 240 ppm of the nanocomposite. The mycelial growth inhibition varies from 100% to 5.14% in different concentrations of prepared nanocomposites (Table 2).

### 3.5. Protein and Isozymes Profile Degradation 

SDS-PAGE analysis was carried out to evaluate the change in gene expression of *A. flavus* handled with a hundred and eighty ppm of Cu-Chit/NCs gel. Some of the protein bands in the *A. flavus* isolates were not seen in the Cu-Chit/NCs gel treatment. In the control group, the protein pattern gave enhanced protein bands. In particular, there were 3 principal bands within the protein maker, inclusive of 66, 45, and 22 kDa, respectively. In high producer (HP) isolate from *A. flavus* treated with 180 ppm of Cu-Chit/NCs gel, five bands completely disappeared with molecular weights of 12, 17, 26, 38, and 55 kDa, respectively. While in intermediate producer (IP) and low producer (LP), isolates were treated with the same concentration of nanocomposites, this resulted in the induction of three newly expressed proteins with approximate molecular weights of 10 kDa, 32 kDa, and 40 kDa (Figure 9A). Native-PAGE results observed that the activity of tested isozymes changed in the treated mycelium. For G6PD activity, six different banding patterns of enzymes appeared in untreated *A. flavus* isolates while enzyme activity in treated isolates with nanocomposites decreased to four isozymes (Figure 9B). Analysis of peroxidase isozymes revealed that from 4–6 peroxidase isozyme loci in both treated and untreated isolates. As a result, Cu-Chit/NCs gels have no effect on peroxidase isozymes activity in fungal mats (Figure 9C). In the present report, we divide *A. flavus* isolates based on their aflatoxins ability into three types: high producer (HP), intermediate producer (IP), and low producer (LP). The same isolates were treated with Cu-Chit/NCs gel as a fellow T1, T2, and T3, respectively.

### 3.6. DNA Binding and Degradation

Separation of genomic DNA of fungi treated with Cu-Chit/NC gel by agarose electrophoresis is broken and DNA band are faint for selected samples, while no serious harm has occurred for untreated DNA. On the other side, the fungi treated with 180 ppm of nanocomposites a slightly less intense band can be observed compared to the untreated sample (Figure 10). The genotoxic effects for fungal mycelium were investigated after treatment with Cu-Chit/NCs gel DNA degradations was separated especially at high concentrations of Cu-Chit /NCs gel.

### 3.7. Fungal SEM

Concerning the morphological structure of *A. flavus* investigated by SEM, adjustments in conidiophore attributes were watched. *A. flavus* was refined on PDA corrected with 180 ppm of Cu-Chit/NCs hydrogel caused slight changes in mycelial structure, highlighted by hyphal twisting and by decreasing of regenerative structures, for example, conidia and conidiophores.

SEM investigation of conidiophore changes showed that fungal spores turned into glaringly extraordinary, in which mycelia and conidiophores were contracted in comparison to untreated controls (Figure 11). Alterations of hyphae structure have been determined as proven in Figure 11A–C along with lower of cytoplasmic content material and adjustments of membrane integrity. whilst, in untreated control, development of mycelium and conidiophore was regular with considerable conidia (Figure 11D–F).

## 4. Discussion

There is an enormous interest to expand effective and eco-friendly fungicides anti-toxigenic fungi with low level or zero mycotoxin residues without affecting the plant growth and crop productivity of the essential agriculture ingredients [9]. The primary target of the current examination is to evaluate the antifungal impacts of bio-polymers like chitosan cross breed with copper metals against *A. flavus* aflatoxin-creating strains. Measures of aflatoxin formed in cultural media show some alarms on the toxigenic capability of various fungal isolates to deliver a high quantity of aflatoxin in agricultural supplies [43]. In the present work, analysis of aflatoxin-producing ability by fluorescence in CA an AFPA medium showed a good correlation with the biochemical examination of aflatoxins, a cutting-edge finding in a harmony with Monda et al. [44]. However, for most purposes, we found the CA media screening technique to be simpler, faster, and much cheaper than any of the different techniques examined [45]. Although the data indicate that the aflatoxins distinguished producer media such as AFPA is not completely persistent in differentiating between aflatoxin-producing and nontoxigenic strains of *A. flavus*, it is important that the fungal medium did not yield false-positives [46].

Our results show that 62% percent of *A. flavus* is aflatoxin-producing isolates. Fifty percent of the screened isolates of *A. flavus* collected from discolored rice grains in India can produce aflatoxin B1 [47]. These results are in agreement with Abbas et al. [48] who investigated more noteworthy producer *A. flavus* strains. Lai et al. [49] indicated that more than 35% of *A. flavus* strains secreted various quantities of aflatoxins in the rice grain. The diverse aflatoxin production capacities of the *A. flavus* isolates would be affected by the various resources of the strains and also ecological problems. The aflatoxin production pathway includes roughly 30 genes, some having unsure functions in aflatoxin biosynthesis [50].

In the present work, *A. flavus* isolates were tested for the presence of gene *alfA*, which is code for fatty acid synthases, while structural gene *aflP* is one of the main genes responsible for transforming ST into O-methylsterigmatocystin [51]. Our findings show that the two primers sets are specific for fast detection of *A. flavus*. Based on specific target genes such as *aflA* and *aflP*, the present findings confirmed the applicability of PCR assays for the detection of *A. flavus* isolated from the feeds. Researchers reported strong antifungal activity of Cu NPs and chitosan nanocomposites against *A. flavus*, for example, benzoic acid nanogel (CS-BA) [10], CuO NPs [12], nanocomposites anti-aflatoxigenic [3,5,13,14]. In the current work, Cu-Chit/NC_S_ hydrogel showed complete inhibition of growth against *A. flavus* strains at the highest concentration (240 ppm). Furthermore, we reported that the antifungal efficacy is influenced not only by nanocomposites concentration but also by type of tested strain. The antifungal efficacy of copper oxide nanoflowers as an antifungal agent against some phytopathogenic fungi like, *A. niger, A. flavus, Penicillium notatum*, and *A. alternata* were reported [52]. In addition, CS-Cu and CSZn NCPs show strong in vitro antifungal activity against *A. alternata, Rhizoctonia solani*, and *A. flavus* and are introduced as potential materials for innovative antimicrobials in cosmetics, foodstuffs, and textiles [53]. In in vitro assays, Cu-chitosan NPs were found to be effective in inhibiting fungal growth of some plant pathogens such as *Alternaria solani* and *Fusarium oxysporum* [54]. The antifungal activity of CS NPs against two aflatoxin producers such as *A. flavus* and *A. parasiticus* was demonstrated [55] and CS NPs succeeded in reducing total aflatoxin production and inhibiting the extent of fungal growth. The main protein composition in the absence or presence of Cu-Chit/NCs gel was analyzed by SDS-PAGE in comparison with protein markers and depending on the amino acid composition. Many protein bands have not been seen in nanocomposite treatment. Subsequently, the treatment of chitosan nanocomposites generates some biological reactions, such as oxidative stress-induced metabolic changes, which in turn affect the protein synthesis rate [56]. The toxicity of nanocomposites in fungal cells is due to severe metabolic changes, in particular protein synthesis, which resulted in a maximum protein reduction, as verified by the absence of the most important protein synthesis [57].

G6PD is a housekeeping enzyme that primarily regenerates adenine dinucleotide phosphate (NADPH) nicotinamide to sustain cellular redox homeostasis. Since NADPH is necessary for NADPH oxidase (NOX), synthase of nitrogen oxides to generate reactive oxygen species, and for signaling nitrogen, several new cellular functions have been established for G6PD [58]. Lack of glucose-6-phosphate dehydrogenase isozyme can make nanoparticles more susceptible to oxidative stress [59]. Several experiments can be checked that they explain the impact of metals on G6PDH activity, also Cd^++^ greatly influences on G6PDH activity in bacteria, fungi, and vertebrates; Ni^++^ inhibits the enzyme ‘s kinetic properties in mammals; Zn^++^ has extreme effects from a variety of influences on G6PDH; Cu^++^ has severe effects from bacteria and animals on G6PDH [60].

G6PD stimulates xenobiotic metabolism via the Nof2 signaling pathway and impacts the xenobiotic-metabolizing expression of the enzyme [61]. The full sense is that the inhibition of *A. aculeatus* G6PD activity by zinc and many other metal nanoparticles may be reinforced by potential production or otherwise formulation of polyketide mycotoxins in toxigenic fungi, including Aspergillus [62]. Additional attempts and modes of action study are needed to examine the molecular mechanisms on which G6PD interacts with the Nrf2 pathway. This is the first report showing G6PD isozymes activity in *A. flavus* strains treated with prepared nanocomposites to understand the antifungal mechanisms. SEM images of the treated pathogen above show that the hyphae also had a swollen appearance, damaging the plasma membrane of both fungal spores and mycelium. Similar outcomes were investigated by Rubina et al. [20], who discovered that Cu-chitosan nanocomposites deteriorate fungal mycelia of *R. solani* from cotton and also *S. rolfsii* pathogenic to onion. Gold nanoparticles may alter and disturb the fungal cell membranes of *A. flavus, F. verticillioides*, and *P. citrinumdue* [63]. Weak sporulation with shrinking spores and defects was found in all *A. versicolor* strains treated with the modified nanocomposites [64]. More omics tools such as functional genomics, transcriptomics, proteomics, and metabolomics are required for the identification of different antifungal mechanism pathways for various nanomaterials that can be used against aflatoxigenic strains of Aspergillus and also suppress their aflatoxins production.

## 5. Conclusions

Dairy cattle feed is prone to fungal infections and major fungus infecting peanut meals, and cotton seeds are *A. flavus* with aflatoxins producing nature. Therefore, an urgent need to produce novel and safer nano-biocides to prevent fungal contamination of food and feed. Current research shows that these two media can only distinguish between aflatoxigenic and non-aflatoxigenic isolates from *A. flavus*. Most *A. flavus* strains react positively with *aflP* primers and *aflA* covering regions 236 bp and 412 bp, respectively. Current results suggest that the prepared nanocomposites hydrogel could be used not only as an effective fungicide against plant pathogens but also can be effectively used for the management of toxigenic fungi. In addition, omics technologies can be extended to improve our knowledge of toxigenic fungi, classify fungal species, predict fungal contamination, and may also facilitate the progress of plant breeding by gene insertion technologies to improve host plant tolerance, deter or minimize contamination of mycotoxins in feed. In addition, omics technologies can be extended to improve our knowledge of toxigenic fungi, classify fungal species, predict fungal contamination, and may also facilitate the progress of plant breeding by nanobiotechnologies to improve host plant tolerance, deter, or minimize contamination of mycotoxins in feed.

## Figures and Tables

**Figure 1 jof-06-00112-f001:**
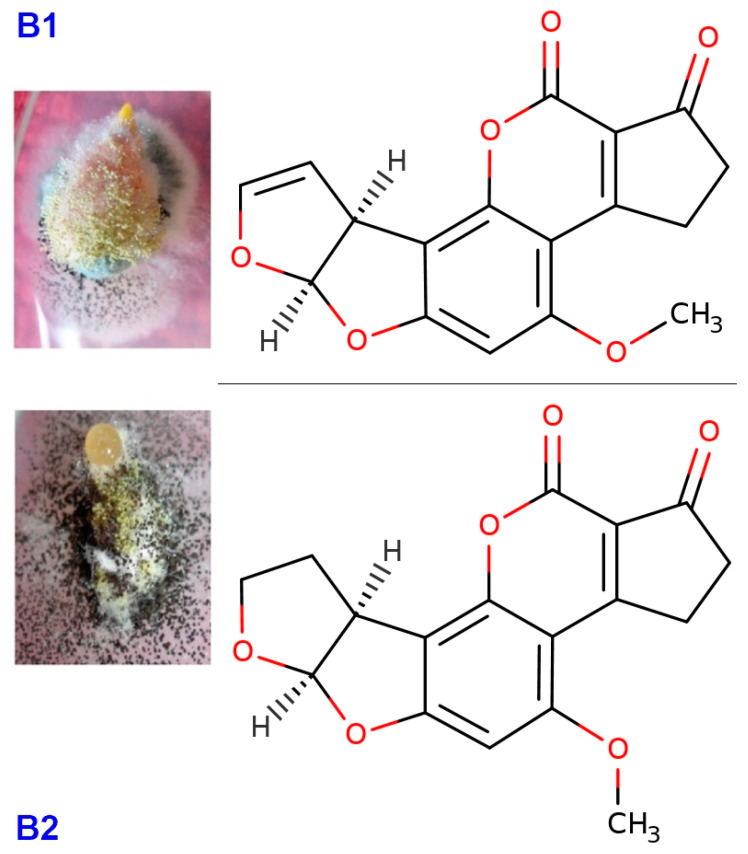
Structural formula of two type of aflatoxins produced by *A. flavus* isolated from peanut meal and cotton seed. Aflatoxin (B1) C_17_H_12_O_6_ (6aR,9aS)-4-Methoxy-2,3,6a,9a-tetrahydrocyclopenta[c]furo[3′,2′:4,5]furo[2,3-h]chromen-1,11-dion. Aflatoxin (B2), C_17_H_14_O_6_ (6aS,9aR)-4-Methoxy-2,3,6a,8,9,9a hexahydrocyclopenta[c]furo[3′,2′:4,5]furo[2,3-h]chromen-1,11-dion. Formula available online from: http://www.chemspider.com.

**Figure 2 jof-06-00112-f002:**
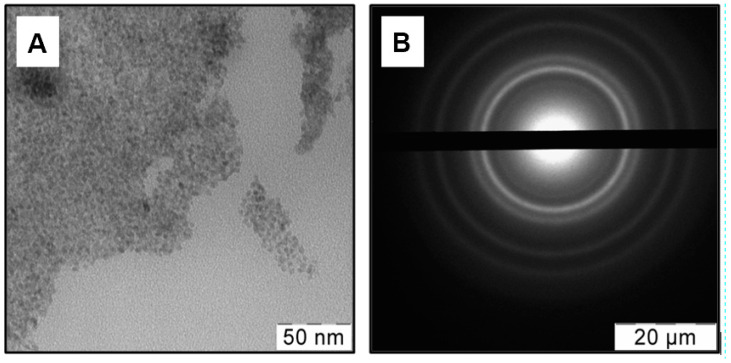
TEM image in bright field (**A**) and selected area diffraction pattern (SAED) (**B**) from a region for Cu-acetone organosol.

**Figure 3 jof-06-00112-f003:**
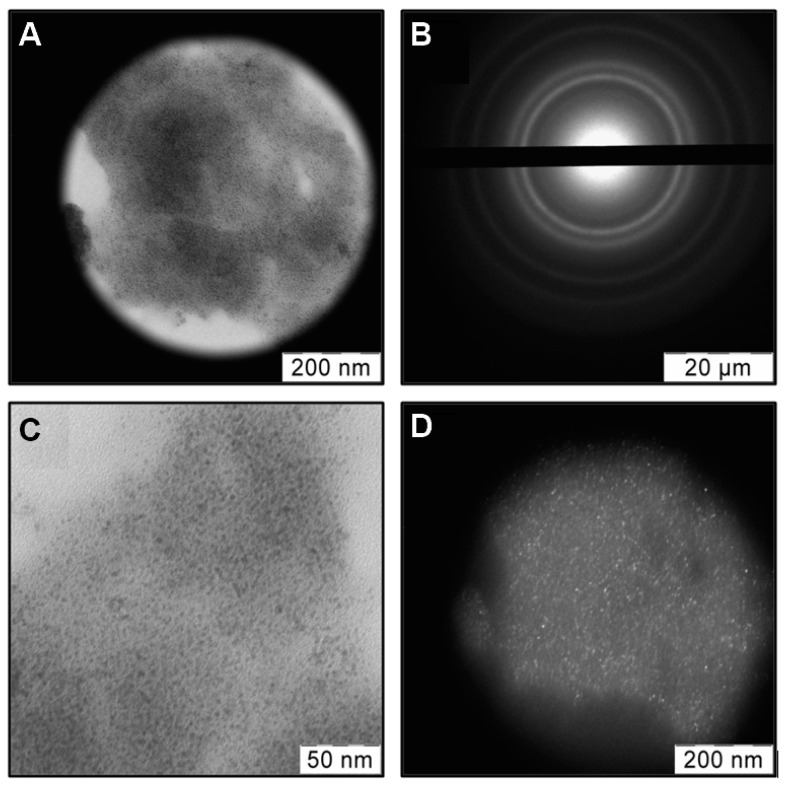
TEM images in bright (**A**,**C**) and dark fields (**D**) of chitosan with a high molecular weight (ChitHMW) doped with copper nanoparticles (Cu NPs) as well as SAED (**B**) of highlighted field.

**Figure 4 jof-06-00112-f004:**
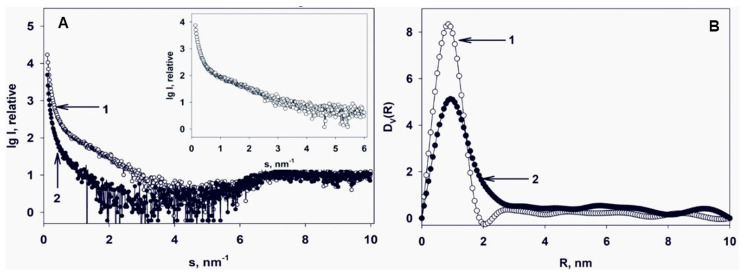
(**A**) Experimental (SAXS) curves: 1—Cu-carrying chitosan; 2—pristine non-modified chitosan (chitosan low molecular weight (ChitLMW)). Insert—difference SAXS curve for the embedded Cu nanoparticles. (**B**) Volume size distribution functions *D_V_(R)*: 1—Cu nanoparticles; 2—pristine non-modified chitosan (ChitLMW).

**Figure 5 jof-06-00112-f005:**
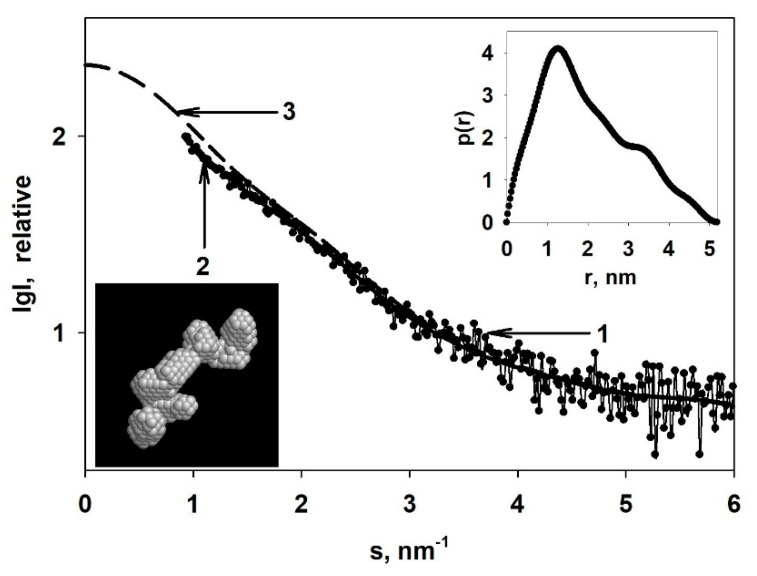
Reconstruction of the shape of the Cu nanoparticles in the Cu-carrying chitosan: 1—difference SAXS curve; 2—a model scattering curve calculated from the restored shape of the Cu nanoparticles; 3—extrapolated to zero angles smoothed scattering curve after the introduction of collimation corrections. Inserts: top right—distance distribution function *p(r)*; bottom left—restored shape of the Cu nanoparticles.

**Figure 6 jof-06-00112-f006:**
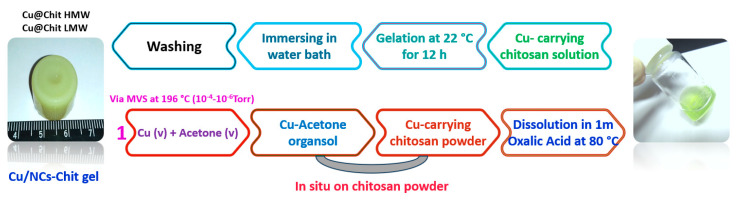
Scheme of preparation of chitosan hydrogels from Cu-carrying chitosan powders.

**Figure 7 jof-06-00112-f007:**
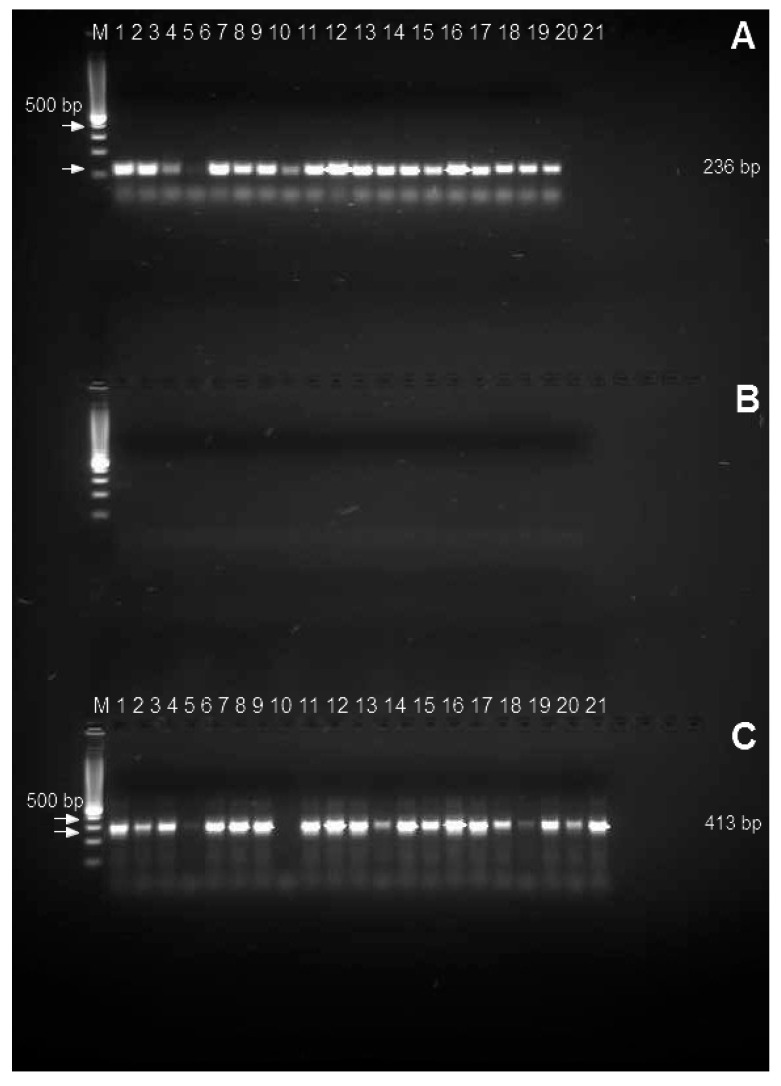
PCR amplicons obtained using primer pairs developed for the *aflP* (*omtA*) and *aflA* genes in 21 *Aspergillus*
*flavus* (Lane1–21) tested with *a**fl**P* primers (**A**), and an *Aspergillus* related isolate such as two isolates for both of *Aspergillus clavatus, Aspergillus ochraceous, Aspergillus niveus, Aspergillus terreus, Aspergillus fumigatus, Aspergillus versicolor, Penicillium paneum*, *Penicillium expansum*, *Penicillium citrinum, Penicillium verrucosum* and one *Alternaria alternate* isolate (Lane 1–21) tested with *a**fl**P* primers (**B**). A total of 21 *Aspergillus*
*flavus* (Lane1–21) tested with *aflA* primers (**C**).

**Figure 8 jof-06-00112-f008:**
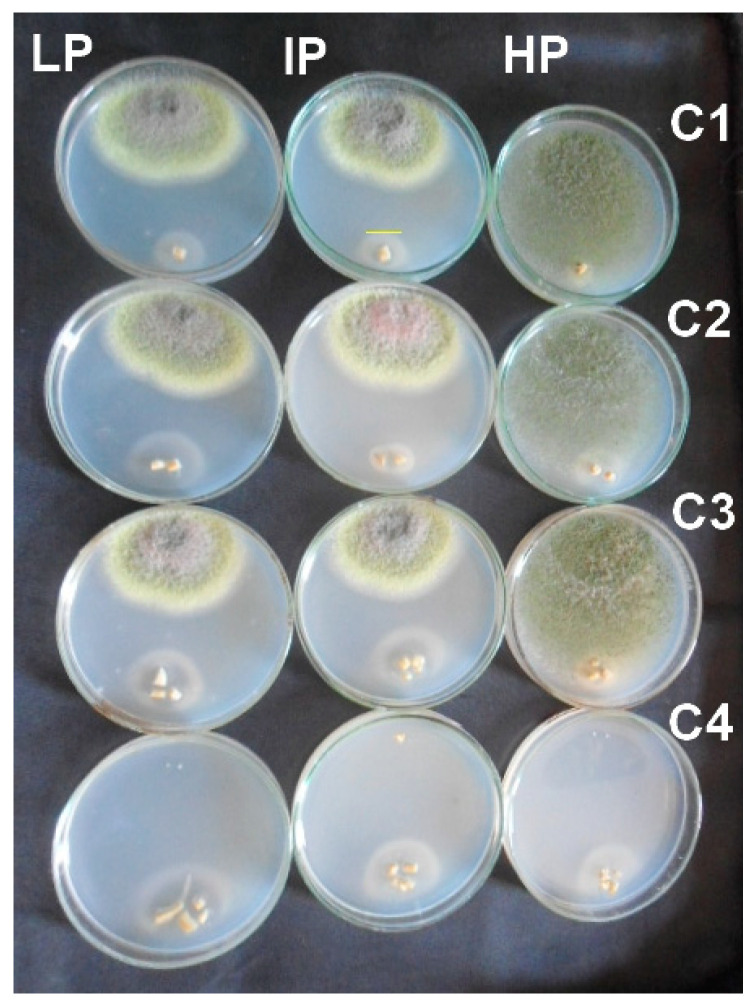
Antifungal activity for different concentration of Cu-Chit/NCs gel (C1 = 60, C2 = 120, C3 = 180, and C4 = 240 ppm) against *A. flavus* isolated from feeds by the plate assay. All Petri dish treatments were incubated at 28 °C for one week.

**Figure 9 jof-06-00112-f009:**
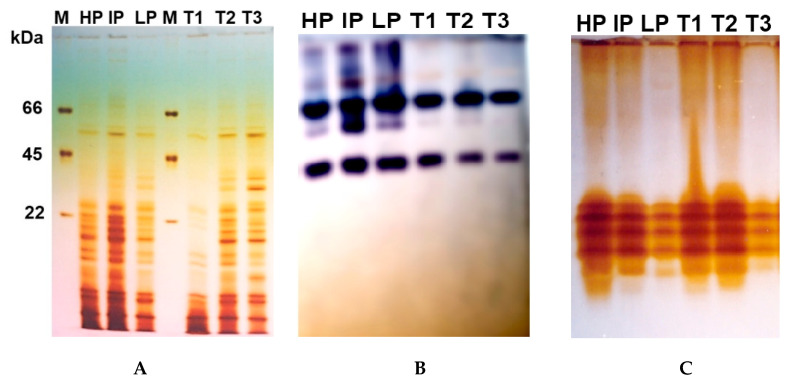
(**A**) Protein expression profile of SDS-PAGE extracted from *A. flavus* mycelium treated with a high Cu-Chit/NCs gel concentration. Lane M shows a standard protein molecular weight marker. Protein marker including three molecular bands ranging from 66, 45, and 22 kDa was used. Isoenzymes electrophoresis of G6PD (**B**) and peroxidase (**C**) isozymes extracted from *A. flavus* mycelium treated with a high Cu-Chit/NCs gel concentration.

**Figure 10 jof-06-00112-f010:**
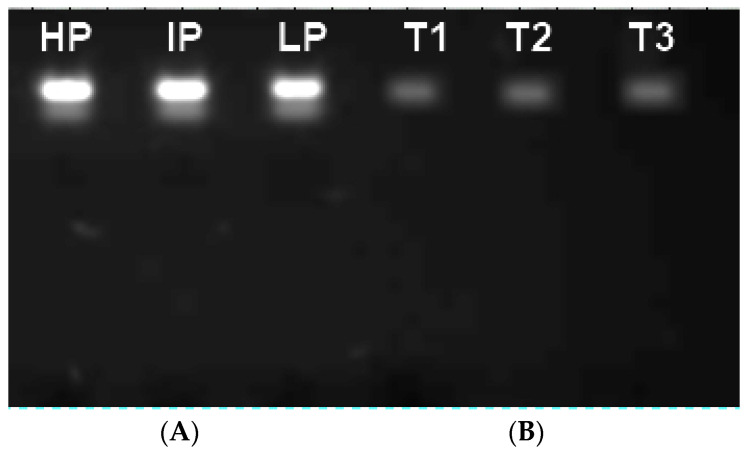
Agarose gel electrophoretic pattern of the fungal genomic DNA treated with 180 ppm of Cu-Chit/NCs gel, (**A**) Lanes 1–3: DNA for untreated *A. flavus* isolates, Lane 1: *A. flavus* (high producer (HP) isolate), Lane 2: *A. flavus* (intermediate producer (IP) isolate), Lane 3: *A. flavus* (low producer (LP) isolate). (**B**) Lanes T1, T2, and T3, three *A. flavus* isolates DNA treated with Cu-Chit/NCs gel, showed total damage to fragmented DNA bands.

**Figure 11 jof-06-00112-f011:**
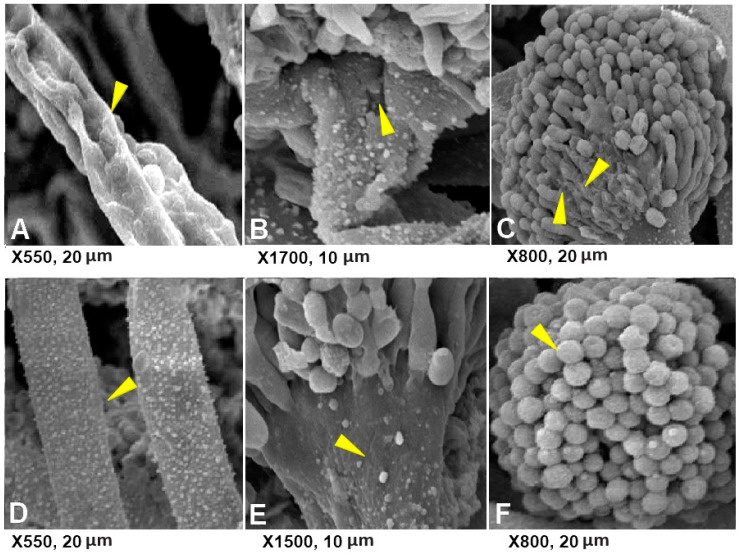
Scanning electron micrographs showing shriveled conidiophores (**A**–**C**) and the healthy mycelium or conidiophores (**D**–**F**) of *A. flavus* on PDA treated with Cu-Chit/NCs hydrogel. The yellow arrows mean hyphae or conidiophore.

**Table 1 jof-06-00112-t001:** Fast screen aflatoxins by various cultural media and concentration of aflatoxins assayed by VICAM test in 21 *Aspergillus flavus* isolates collected from peanut meal and cottonseeds.

Isolate Code	Feed	Fluorescence Detection under UV Light (365 nm)				The Concentration of Aflatoxins (ppb)		
		AFPA	CA	PDA + Na Cl	PDA	AFB1	AFB2	Total Aflatoxins
Af1	Peanut meal	+	−	+	−	19.44	0.03	19.47
Af2	Peanut meal	+	+	+	−	13.54	0.02	13.56
Af3	Peanut meal	+	+	−	−	10.13	0.05	10.18
Af4	Peanut meal	+	+	+	−	14.22	5.29	19.51
Af5	Peanut meal	+	−	−	−	10.14	1.30	11.44
Af6	Peanut meal	+	+	−	−	ND	ND	ND
Af7	Peanut meal	−	+	+	−	12.10	3.20	15.30
Af8*	Peanut meal	+	+	+	−	12.10	3.56	15.66
Af9	Peanut meal	+	+	−	−	ND	ND	ND
Af10	Peanut meal	+	+	+	−	10.13	0.05	10.18
Af11	Cotton seeds	+	−	−	−	ND	ND	ND
Af12	Cotton seeds	+	+	+	−	7.20	2.13	9.33
Af13	Cotton seeds	+	+	+	−	9.10	2.50	11.60
AF14	Cotton seeds	+	+	+	−	8.14	1.34	9.48
AF15	Cotton seeds	+	−	+	−	ND	ND	ND
AF16	Cotton seeds	+	+	+	−	ND	ND	ND
AF17	Cotton seeds	+	+	−	−	10.64	1.36	12.00
AF18	Cotton seeds	−	+	+	−	ND	ND	ND
AF19	Cotton seeds	+	+	+	−	4.50	1.30	5.80
AF20	Cotton seeds	+	−	−	−	ND	ND	ND
AF21	Cotton seeds	−	−	−	−	ND	ND	ND

AF: Aflatoxin, +: Positive fluorescence, −: Negative fluorescence, ND: Not detected, *Aspergillus flavus* and *parasiticus* Agar (AFPA), Coconut agar (CA), potato dextrose agar (PDA), *Af4, highly producer isolate. Af13, Intermediate producer isolate. Af19, Low producer isolate.

**Table 2 jof-06-00112-t002:** Antifungal activity of Cu-Chit/NCs gel against three *Aspergillus flavus* strains.

Serial Number	NCs Gel Concentrations (ppm)	Strain Code	% Inhibition
1	C1 = 60	LP	27.89
2		IP	28.66
3		HP	5.14
4	C2 = 120	LP	25.67
5		IP	28.43
9		HP	6.48
7	C3 = 180	LP	29.75
8		IP	29.20
9		HP	6.57
10	C4 = 240	LP	100
11		IP	100
12		HP	100

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
