# Peer review of "Copper-Chitosan Nanocomposite Hydrogels Against Aflatoxigenic Aspergillus flavus from Dairy Cattle Feed"

_jof, 2020, doi:10.3390/jof6030112_

Round 1

Reviewer 1 Report

The authors should check the stereochemistry of Aflatoxin B1 and Aflatoxin B2 in figure 1 is correct.    The stereochemistry in the figures does not match some widely accessible figures of these aflatoxins.  Usually the hydrogens are drawn near the furan-type ring to communicate the stereochemistry.  The authors could add these two hydrogens to make the stereochemistry clear.  The stereochemistry of aflatoxin B2 does not match other reports in the literature.

Author Response

1-Based on reviewer comment, we modified Figure 1, see the revised version.

Reviewer 2 Report

The study is very complex and interesting. I suggest few minor corrections.

  1. As a general remark- names of genera and species should be written in italic, all over the text
  2. pag 2 line 68-69. Niger, A. The flavus showed.... is unclair, rephrase
  3. pag 2line 78 in order to prevent management aflatoxicosis in dairy cow= in order to prevent aflatoxicosis in dairy cow
  4. pag 2 line 80 from animal feed samples is not thoroughly studied. Make a conexion with the next sentences, Eg: The study aimed to: 1)....
  5. pag 3, line 123 chitosan gels were stored in a water/isopropanol (6:1, v/v) bath at room temperature (RT).
  6. pag 8 line 307 - Legend of Fig 4 should be placed entirely under the Figure.
  7. pag 15, line 471 (Fig.)..do mention which Fig

Author Response

As a general remark- names of genera and species should be written in italic, all over the text

We checked and modified all microbes names

pag 2 line 68-69. Niger, A. The flavus showed.... is unclair, rephrase

Re-phrased

pag 2line 78 in order to prevent management aflatoxicosis in dairy cow= in order to prevent aflatoxicosis in dairy cow

Modified

pag 2 line 80 from animal feed samples is not thoroughly studied. Make a conexion with the next sentences, Eg: The study aimed to: 1)....

Modified

pag 3, line 123 chitosan gels were stored in a water/isopropanol (6:1, v/v) bath at room temperature (RT).

Modified

pag 8 line 307 - Legend of Fig 4 should be placed entirely under the Figure.

Transfered

pag 15, line 471 (Fig.)..do mention which Fig.

Inserted